# 3D Pose Transfer with Correspondence Learning and Mesh Refinement

Chaoyue Song[1], Jiacheng Wei[2], Ruibo Li[1,3], Fayao Liu[4] and Guosheng Lin[1,3*]

[1]S-Lab, Nanyang Technological University, Singapore
[2]School of Electrical and Electronic Engineering, Nanyang Technological University, Singapore
[3]School of Computer Science and Engineering, Nanyang Technological University, Singapore
[4]Institute for Inforcomm Research, A*STAR, Singapore
{chaoyue.song, gslin}@ntu.edu.sg

## Abstract

3D pose transfer is one of the most challenging 3D generation tasks. It aims to transfer the pose of a source mesh to a target mesh and keep the identity (e.g., body shape) of the target mesh. Some previous works require key point annotations to build reliable correspondence between the source and target meshes, while other methods do not consider any shape correspondence between sources and targets, which leads to limited generation quality. In this work, we propose a correspondence-refinement network to achieve the 3D pose transfer for both human and animal meshes. The correspondence between source and target meshes is first established by solving an optimal transport problem. Then, we warp the source mesh according to the dense correspondence and obtain a coarse warped mesh. The warped mesh will be better refined with our proposed *Elastic Instance Normalization*, which is a conditional normalization layer and can help to generate high-quality meshes. Extensive experimental results show that the proposed architecture can effectively transfer the poses from source to target meshes and produce better results with satisfied visual performance than state-of-the-art methods. Our code and data are available at https://github.com/ChaoyueSong/3d-corenet.

## 1 Introduction

3D pose transfer has been drawing a lot of attention from the vision and graphics community. It has potential applications in 3D animated movies and games by generating new poses for existing shapes and animation sequences. 3D pose transfer is a learning-driven generation task which is similar to style transfer on 2D images. As shown in Figure 1, pose transfer takes two inputs, one is identity mesh that provides mesh identity (e.g., body shape), the other is pose mesh that provides the pose of mesh. It aims at transferring the pose of a source pose mesh to a target identity mesh and keeping the identity of the target identity mesh.

A fundamental problem for previous methods is to build reliable correspondence between source and target meshes. It can be very challenging when the source and target meshes have significant differences. Most of the previous methods try to solve it with the help of user effort or other additional inputs, such as key point annotations [3, 34, 41], etc. Unfortunately, it is time-consuming to obtain such additional inputs that will limit the usage in practice. In [37], they proposed to implement pose transfer without correspondence learning. Their method is convenient but the performance will be degraded since they do not consider the correspondence between meshes. In this work, we propose a *COrrespondence-REfinement Network (3D-CoreNet)* to solve the pose transfer problem for both the human and animal meshes. Like [37], our method does not need key point annotations or other

---

*Corresponding author

35th Conference on Neural Information Processing Systems (NeurIPS 2021).

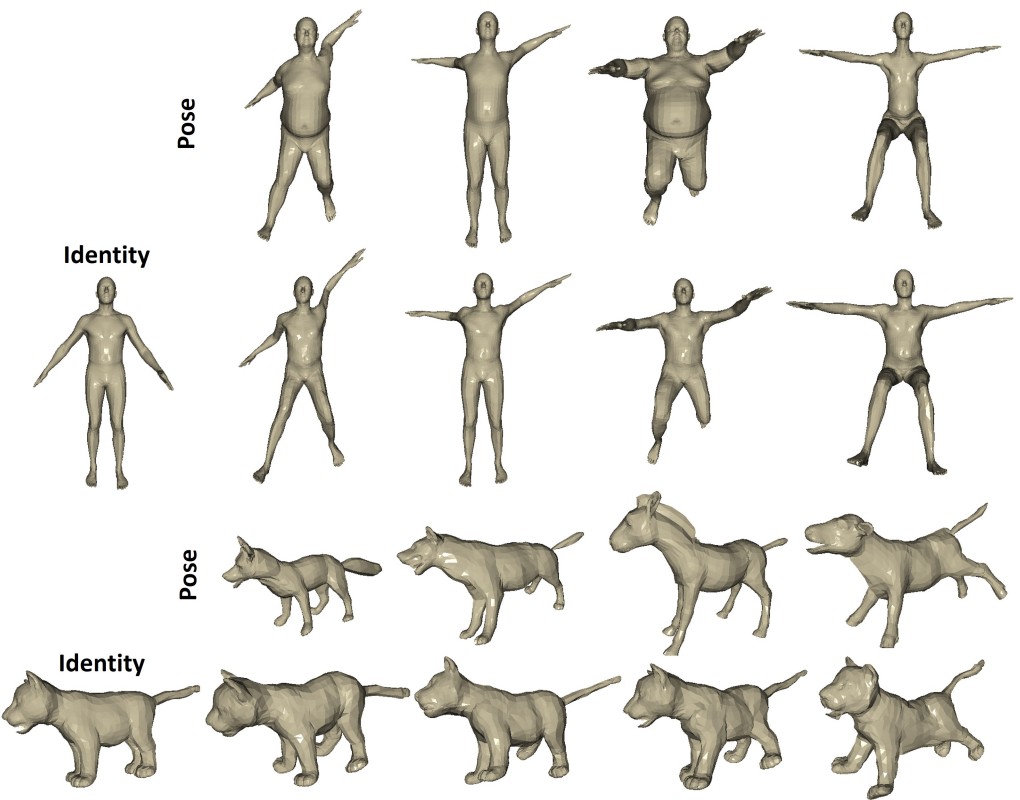

Figure 1: **Pose transfer results generated by our *3D-CoreNet*.** In the first two rows, the human identity and pose meshes are from SMPL [23], in the last two rows, the animal identity and pose meshes are from SMAL [46].

additional inputs. We learn the shape correspondence between identity and pose meshes first, then we warp the pose mesh to a coarse warped output according to the correspondence. Finally, the warped mesh will be refined to have a better visual performance. Our method does not require the two meshes to have the same number or order of vertices.

For the correspondence learning module, we treat the shape correspondence learning as an optimal transport problem to learn the correspondence between meshes. Our network takes vertex coordinates of identity and pose meshes as inputs. We extract deep features at each vertex using point cloud convolutions and compute a matching cost between the vertex sets with the extracted features. Our goal is to minimize the matching cost to get an optimal matching matrix. With the optimal matching matrix, we warp the pose mesh and obtain a coarse warped mesh. We then refine the warped output with a set of elastic instance normalization residual blocks, the modulation parameters in the normalization layers are learned with our proposed *Elastic Instance Normalization (ElaIN)*. In order to generate smoother meshes with more details, we introduce a channel-wise weight in ElaIN to adaptively blend feature statistics of original features and the learned parameters from external data, which help to keep the consistency and continuity of the original features.

Our contributions can be summarized as follows:
• We solve the 3D pose transfer problem with our proposed correspondence-refinement network. To the best of our knowledge, our method is the first to learn the correspondence between different meshes and refine the generated meshes jointly in the 3D pose transfer task.
• We learn the shape correspondence by solving an optimal transport problem without any key point annotations and generate high-quality final meshes with our proposed elastic instance normalization in the refinement module.
• Through extensive experiments, we demonstrate that our method outperforms state-of-the-art methods quantitatively and qualitatively on both human and animal meshes.

## 2 Related work

### 2.1 Deep learning methods on 3D data

The representations of 3D data are various, like point clouds, voxels and meshes. 3DShapeNets [40] and VoxNet [25] propose to learn on volumetric grids. Their methods cannot be applied on complex data due to the sparsity of data and computation cost of 3D convolution. PointNet [29] uses a shared MLP on every point followed by a global max-pooling. Following PointNet, some hierarchical architectures have been proposed to aggregate local neighborhood information with MLPs [20, 30]. [13, 35] proposed mesh variational autoencoder to learn mesh features whose methods consume a large amount of computing resources due to their fully-connected networks. Many works use graph convolutions with mesh down- and up-sampling layers [15], like CoMA [31], CAPE [24] based on ChebyNet [10], and [45] based on SpiralNet [21], SpiralNet++ [16], etc. They all need a template to implement their hierarchical structure which is not applicable to real-world problems. In this work, we use mesh as the representation of 3D shape and shared weights convolution layers in the network.

### 2.2 3D pose transfer

Deformation transfer in graphics aims to apply the deformation exhibited by a source mesh onto a different target mesh [34]. 3D pose transfer aims to generate a new mesh based on the knowledge of a pair of source and target meshes. In [3, 34, 41, 42], the methods all require to label the corresponding landmarks first to deal with the differences between meshes. Baran et al. [2] proposed a method that infers a semantic correspondence between different poses of two characters with the guidance of example mesh pairs. Chu et al. [8] proposed to use a few examples to generate results which will make it difficult to automatically transfer pose. For this problem, Gao et al. [14] proposed to use cycle consistency to achieve the pose transfer. However, their method cannot deal with new identities due to the limitations of the visual similarity metric. In [37], they solved the pose transfer via the latest technique for image style transfer. Their work does not need other guidance, but the performance is also restrained since they do not learn any correspondence. To solve the problems, our network learns the correspondence and refines the generated meshes jointly.

### 2.3 Correspondence learning

In CoCosNet [44], they introduced a correspondence network based on the correlation matrix between images without any constraints. To learn a better matching, we proposed to use optimal transport to learn the correspondence between meshes. Recently, optimal transport has received great attention in various computer vision tasks. Courty et al. [9] perform the alignment of the representations in the source and target domains by learning a transportation plan. Su et al. [33] compute the optimal transport map to deal with the surface registration and shape space problem. Other applications include generative model [1, 6, 11, 39], scene flow [28], semantic correspondence [22] and etc.

### 2.4 Conditional normalization layers

After normalizing the activation value, conditional normalization uses the modulation parameters calculated from the external data to denormalize it. Adaptive Instance Normalization (AdaIN) [18] aligns the mean and variance between content and style image which achieves arbitrary style transfer. Soft-AdaIN [7] introduces a channel-wise weight to blend feature statistics of content and style image to preserve more details for the results. Spatially-Adaptive Normalization (SPADE) [27] can better preserve semantic information by not washing away it when applied to segmentation masks. SPAdaIN [37] changes batch normalization [19] in SPADE to instance normalization [36] for 3D pose transfer. However, it will break the consistency and continuity of the feature map when doing the denormalization, which has a bad influence on the mesh smoothness and detail preservation. To address this problem, our ElaIN introduces an adaptive weight to implement the denormalization.

## 3 Method

Given a source pose mesh and a target identity mesh, our goal is to transfer the pose of source mesh to the target mesh and keep the identity of the target mesh. In this section, we will introduce our end-to-end *Correspondence-Refinement Network (3D-CoreNet)* for 3D pose transfer.

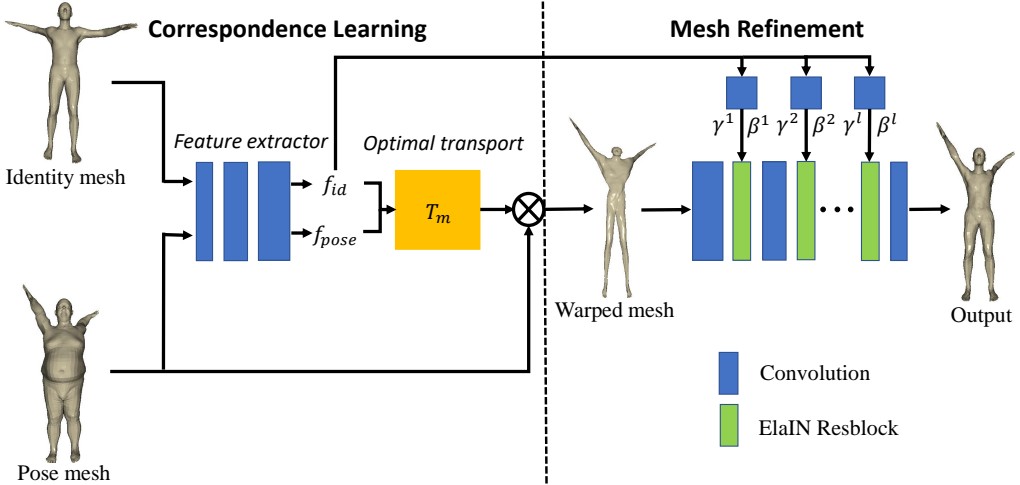

Figure 2: **The architecture of *3D-CoreNet*.** With the extracted features, the shape correspondence between identity and pose meshes is first established by solving an optimal transport problem. Then, we warp the pose mesh according to the optimal matching matrix $\mathbf{T}_m$ and obtain a coarse warped mesh. The warped mesh will be better refined with our proposed ElaIN in the refinement module.

A 3D mesh can be represented as $M(I, P, O)$, where $I$ denotes the mesh identity, $P$ represents the pose of the mesh, $O$ is the vertex order of the mesh. Given two meshes $M_{id} = M(I_1, P_1, O_1)$ and $M_{pose} = M(I_2, P_2, O_2)$, we aim to transfer the pose of $M_{pose}$ to $M_{id}$ and generate the output mesh $M_{output} = M'(I_1, P_2, O_1)$. In our *3D-CoreNet*, we take $V_{id} \in \mathbb{R}^{N_{id} \times 3}$ and $V_{pose} \in \mathbb{R}^{N_{pose} \times 3}$ as inputs, which are the $(x, y, z)$ coordinates of the mesh vertices. $N_{id}$ and $N_{pose}$ denote the number of vertices of identity mesh and pose mesh respectively.

As shown in Figure 2, the vertices of the meshes are first fed into the network to extract multi-scale deep features. We calculate the optimal matching matrix with the vertex feature maps by solving an optimal transport problem, then we warp the pose mesh according to the matrix and obtain the warped mesh. Finally, the warped mesh is refined to the final output mesh with our proposed elastic instance normalization (ElaIN). The output mesh combines the pose from the source mesh and the identity from the target. And it inherits the vertex order from the identity mesh.

## 3.1 Correspondence learning

Given an identity mesh and a pose mesh, our correspondence learning module calculates an optimal matching matrix, each element of the matching matrix represents the similarities between two vertices in the two meshes. The first step in our shape correspondence learning is to compute a correlation matrix with the extracted features, which is based on cosine similarity and denotes the matching similarities between any two positions from different meshes. However, the matching scores in the correlation matrix are calculated without any additional constraints. To learn a better matching, we solve this problem from a global perspective by modeling it as an optimal transport problem.

**Correlation matrix** We first introduce our feature extractor which aims to extract features for the unordered input vertices. Close to [37], our feature extractor consists of 3 stacked $1 \times 1$ convolution and Instance Normalization layers, the activation functions applied are all LeakyReLU. Given the extracted vertex feature maps $f_{id} \in \mathbb{R}^{D \times N_{id}}$, $f_{pose} \in \mathbb{R}^{D \times N_{pose}}$ of identity and pose meshes ($D$ is the channel-wise dimension), a popular method to compute correlation matrix is using the cosine similarity [43, 44]. Concretely, we compute the correlation matrix $\mathbf{C} \in \mathbb{R}^{N_{id} \times N_{pose}}$ as:

$$\mathbf{C}(i, j) = \frac{f_{id}(i)^\top f_{pose}(j)}{\|f_{id}(i)\| \, \|f_{pose}(j)\|} \tag{1}$$

where $\mathbf{C}(i, j)$ denotes the individual matching score between $f_{id}(i)$ and $f_{pose}(j) \in \mathbb{R}^{\mathbb{D}}$, $f_{id}(i)$ and $f_{pose}(j)$ represent the channel-wise feature of $f_{id}$ at position i and $f_{pose}$ at j.

**Optimal transport problem**   To learn a better matching with additional constraints in this work, we model our shape correspondence learning as an optimal transport problem. We first define a matching matrix $\mathbf{T} \in \mathbb{R}_+^{N_{id} \times N_{pose}}$ between identity and pose meshes. Then we can get the total correlation as $\sum_{ij} \mathbf{C}(i,j)\mathbf{T}(i,j)$. The aim will be maximizing the total correlation score to get an optimal matching matrix $\mathbf{T}_m$.

We treat the correspondence learning between identity and pose meshes as the transport of mass. A mass which is equal to $N_{id}^{-1}$ will be assigned to each vertex in the identity mesh, and each vertex in pose mesh will receive the mass $N_{pose}^{-1}$ from identity mesh through the built correspondence between vertices. Then if we define $\mathbf{Z} = 1 - \mathbf{C}$ as the cost matrix, our goal can be formulated as a standard optimal transport problem by minimizing the total matching cost,

$$\mathbf{T}_m = \underset{\mathbf{T} \in \mathbb{R}_+^{N_{id} \times N_{pose}}}{\arg\min} \sum_{ij} \mathbf{Z}(i,j)\mathbf{T}(i,j) \quad s.t. \quad \mathbf{T}\mathbf{1}_{N_{pose}} = \mathbf{1}_{N_{id}}N_{id}^{-1}, \quad \mathbf{T}^\top \mathbf{1}_{N_{id}} = \mathbf{1}_{N_{pose}}N_{pose}^{-1}. \tag{2}$$

where $\mathbf{1}_{N_{id}} \in \mathbb{R}^{N_{id}}$ and $\mathbf{1}_{N_{pose}} \in \mathbb{R}^{N_{pose}}$ are vectors whose elements are all $1$. The first constraint in Eq. 2 means that the mass of each vertex in $M_{id}$ will be entirely transported to some of the vertices in $M_{pose}$. And each vertex in $M_{pose}$ will receive a mass $N_{pose}^{-1}$ from some of the vertices in $M_{id}$ with the second constraint. This problem can be solved by the Sinkhorn-Knopp algorithm [32]. The details of the solved process will be given in the supplementary material.

With the matching matrix, we can warp the pose mesh and obtain the vertex coordinates $V_{warp} \in \mathbb{R}^{N_{id} \times 3}$ of the warped mesh,

$$V_{warp}(i) = \sum_j \mathbf{T}_m(i,j)V_{pose}(j) \tag{3}$$

the warped mesh $M_{warp}$ inherits the number and order of vertex from identity mesh and can be reconstructed with the face information of identity mesh as shown in Figure 2.

## 3.2   Mesh refinement

In this section, we introduce our mesh refinement module which refines the warped mesh to the desired output progressively.

**Elastic instance normalization**   Previous conditional normalization layers [18, 27, 37] used in different tasks always calculated their denormalization parameters only with the external data. We argue that it may break the consistency and continuity of the original features. Inspired by [7], we propose *Elastic Instance Normalization (ElaIN)* which blends the statistics of original features and the learned parameters from external data adaptively and elastically.

As shown in Figure 2, the warped mesh is flatter than we desired and is kind of out of shape, but it inherits the pose from the source mesh successfully. Therefore, we refine the warped mesh with the identity feature maps to get a better final output. Here, we let $h^i \in \mathbb{R}^{S^i \times D^i \times N^i}$ as the activation value before the $i$-th normalization layer, where $S^i$ is the batch size, $D^i$ is the dimension of feature channel and $N^i$ is the number of vertex. At first, we normalize the feature maps of the warped mesh with instance normalization and the mean and standard deviation are calculated across spatial dimension ($n \in N^i$) for each sample $s \in S^i$ and each channel $d \in D^i$,

$$\mu^i = \frac{1}{N^i} \sum_n h^i_{warp} \tag{4}$$

$$\sigma^i = \sqrt{\frac{1}{N^i} \sum_n (h^i_{warp} - \mu^i)^2 + \epsilon} \tag{5}$$

then the feature maps of the identity mesh are fed into a $1 \times 1$ convolution layer to get $h^i_{id}$, which shares the same size with $h^i_{warp}$. We calculate the mean of $h^i_{warp}$, $h^i_{id}$ to make them $S^i \times D^i \times 1$ tensors. The tensors are then concatenated in channel dimension to get a $S^i \times (2D^i) \times 1$ tensor.

A fully-connected layer is employed to compute an adaptive weight $w(h^i_{warp}, h^i_{id}) \in \mathbb{R}^{S^i \times D^i \times 1}$ with the concatenated tensor. With $w(h^i_{warp}, h^i_{id})$, we can define the modulation parameters of our normalization layer.

$$\begin{aligned}
\gamma'(h^i_{warp}, h^i_{id}) &= w(h^i_{warp}, h^i_{id})\gamma^i + (1 - w(h^i_{warp}, h^i_{id}))\sigma^i, \\
\beta'(h^i_{warp}, h^i_{id}) &= w(h^i_{warp}, h^i_{id})\beta^i + (1 - w(h^i_{warp}, h^i_{id}))\mu^i.
\end{aligned} \tag{6}$$

where $\gamma^i$ and $\beta^i$ are learned from the identity feature $h^i_{id}$ with two convolution layers. Finally, we can scale the normalized $h^i_{warp}$ with $\gamma'$ and shift it with $\beta'$,

$$\text{ElaIN}(h^i_{warp}, h^i_{id}) = \gamma'(h^i_{warp}, h^i_{id})(\frac{h^i_{warp} - \mu^i}{\sigma^i}) + \beta'(h^i_{warp}, h^i_{id}) \tag{7}$$

**Mesh refinement module**    Our mesh refinement module is designed to refine the warped mesh progressively. Following [27, 37, 44], We design the ElaIN residual block with our ElaIN in the form of ResNet blocks [17]. As shown in Figure 2, our architecture contains $l$ ElaIN residual blocks. Each of them consists of our proposed ElaIN followed by a simple convolution layer and LeakyReLU. With the ElaIN residual blocks, the warped mesh is refined to our desired high-quality output. Please refer to the supplementary material for the detailed architecture of ElaIN.

### 3.3   Loss function

We jointly train the correspondence learning module along with mesh refinement module by minimizing the following loss functions,

**Reconstruction loss**    Following [37], we train our network with the supervision of the ground truth mesh $M_{gt} = M(I_1, P_2, O_1)$. We first process the ground truth mesh to have the same vertex order as the identity mesh. Then we define the reconstruction loss by calculating the point-wise $L2$ distance between the vertices of $M_{output}$ and $M_{gt}$,

$$\mathcal{L}_{rec} = \|V_{output} - V_{gt}\|^2_2 \tag{8}$$

where $V_{output}$ and $V_{gt} \in \mathbb{R}^{N_{id} \times 3}$ are the vertices of $M_{output}$ and $M_{gt}$ respectively. Notice that they all share the same size and order with the vertices of the identity mesh. With the reconstruction loss, the mesh predicted by our model will be closer to the ground truth.

**Edge loss**    In this work, we also introduce edge loss which is often used in 3D mesh generation tasks [26, 37, 38]. Since the reconstruction loss does not consider the connectivity of mesh vertices, the generated mesh may suffer from flying vertices and overlong edges. Edge loss can help penalize flying vertices and generate smoother surfaces. For every $v \in V_{output}$, let $\mathcal{N}(v)$ be the neighbor of $v$, the edge loss can be defined as,

$$\mathcal{L}_{edg} = \sum_v \sum_{p \in \mathcal{N}(v)} \|v - p\|^2_2 \tag{9}$$

then we can train our network with the combined loss function $\mathcal{L}$,

$$\mathcal{L} = \lambda_{rec}\mathcal{L}_{rec} + \mathcal{L}_{edg} \tag{10}$$

where $\lambda_{rec}$ denotes the weight of reconstruction loss.

## 4   Experiment

**Datasets.**    For the human mesh dataset, we use the same dataset generated by SMPL [23] as [37]. This dataset consists of 30 identities with 800 poses. Each mesh has 6890 vertices. For the training data, we randomly choose 4000 pairs (identity and pose meshes) from 16 identities with 400 poses and shuffle them every epoch. The ground truth meshes will be determined according to the identity and pose parameters from the pairs. Before feeding into our network, every mesh will be shuffled randomly to be close to the real-world problem. Notice that the ground truth mesh will share the

Table 1: **Quantitative comparison with other methods.** We compare our method with DT (needs key point annotations) and NPT using PMD, CD, EMD as our evaluation metrics on both human and animal data. For them, the lower is better. The PMD and CD are in units of $10^{-3}$ and the EMD is in units of $10^{-2}$.

|  | Annotation | Dataset | PMD | CD | EMD |
|---|---|---|---|---|---|
| DT [34] | Key points |  | 0.15 | 0.35 | 2.21 |
| NPT [37] | - | SMPL [23] | 0.66 | 1.42 | 4.22 |
| Ours | - |  | **0.08** | **0.22** | **1.89** |
| DT [34] | Key points |  | 13.37 | 35.77 | 15.90 |
| NPT [37] | - | SMAL [46] | 6.75 | 14.52 | 11.65 |
| Ours | - |  | **2.26** | **4.05** | **7.28** |

same vertex order with identity mesh for the convenience of supervised training and evaluation. For all input meshes, we shift them to the center according to their bounding boxes. When doing the test, we evaluate our model with 14 new identities with 200 unseen poses. We randomly choose 400 pairs for testing. They will be pre-processed in the same manner as the training data. To further evaluate the generalization of our model, we also test our model with FAUST [5] and MG-dataset [4] in the experiment.

For the animal mesh dataset, we generate animal training and test data using SMAL model [46]. This dataset has 41 identities with 600 poses. The 41 identities are 21 felidae animals (1 cat, 5 cheetahs, 8 lions, 7 tigers), 5 canidae animals (2 dogs, 1 fox, 1 wolf, 1 hyena), 8 equidae animals (1 deer, 1 horse, 6 zebras), 4 bovidae animals (4 cows), 3 hippopotamidae animals (3 hippos). Every mesh has 3889 vertices. For the training data, we randomly choose 11600 pairs from 29 identities (16 felidae animals, 3 canidae animals, 6 equidae animals, 2 bovidae animals, 2 hippopotamidae animals) with 400 poses. For the test data, we randomly choose 400 pairs from other 12 identities (5 felidae animals, 2 canidae animals, 2 equidae animals, 2 bovidae animals, 1 hippopotamidae animal) with 200 poses. All the inputs are pre-processed in the same manner as we do in the human mesh.

**Evaluation metrics.** Following [37], we use Point-wise Mesh Euclidean Distance (PMD) as one of our evaluation metrics. PMD is the $L2$ distance between the vertices of the output mesh and the ground truth mesh. We also evaluate our model with Chamfer Distance (CD) and Earth Mover's Distance (EMD) proposed in [12]. For PMD, CD and EMD, the lower is better.

**Implementation details.** $\lambda_{rec}$ in the loss function is set as 2000. We implement our model with Pytorch and use Adam optimizer. Please refer to the supplementary material for the details of the network. Our model is trained for 200 epochs on one RTX 3090 GPU, the learning rate is fixed at $1e$-$4$ in the first 100 epochs and decays $1e$-$6$ each epoch after 100 epochs. The batch size is 8.

## 4.1 Comparison with the state-of-the-arts

In this section, we compare our method with Deformation Transfer (DT) [34] and Neural Pose Transfer (NPT) [37]. DT needs to rely on the corresponding points labeled by user and a reference mesh, as the additional inputs. Therefore, we test DT with the reference mesh and 11, 19 labeling points on animal data and human data respectively. For [37], their method does not consider any correspondence. We train their model using the implementations provided by the authors. The qualitative results tested on SMPL [23] and SMAL [46] are shown in Figure 3 and Figure 4. As we can see, when tested on human data, DT and our method can produce better results that are close to the ground truth. However, it is very time-consuming for DT when dealing with a new identity and pose mesh pair. The results generated by NPT always do not learn the right pose and are not very smooth on the arms or legs since they do not consider any correspondence. When tested on animal data, DT fails to transfer the pose even if we add more labeling points. Their method does not work when the identity of the mesh pairs are very different. NPT can produce very flat legs and wrong direction faces. In comparison, our method still produces satisfactory results efficiently.

We adopt Point-wise Mesh Euclidean Distance (PMD), Chamfer Distance (CD) and Earth Mover's Distance (EMD) to evaluate the generated results of different methods. All metrics are calculated

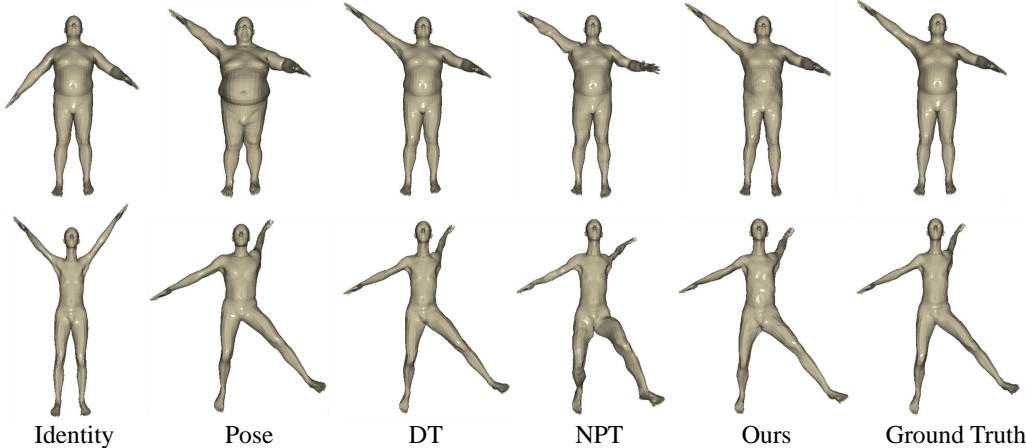

| Identity | Pose | DT | NPT | Ours | Ground Truth |

Figure 3: **Qualitative comparison of different methods on human data.** The identity and pose meshes are from SMPL [23]. Our method and DT (needs key point annotations) can generate better results than NPT when doing pose transfer on human meshes. The results generated by NPT are always not smooth on the arms or legs. Since DT needs user to label the key point annotations, our method is more efficient and practical than DT.

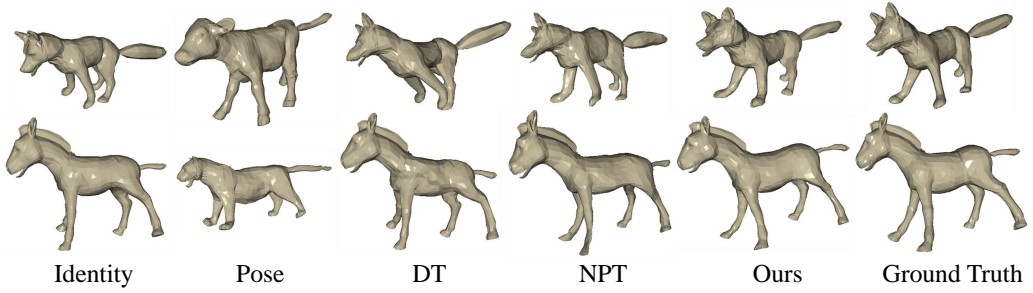

| Identity | Pose | DT | NPT | Ours | Ground Truth |

Figure 4: **Qualitative comparison of different methods on animal data.** The identity and pose meshes are from SMAL [46]. Our method produces more successful results when doing pose transfer on different animal meshes. Although DT has key point annotations, it still fails to transfer the pose when the identity of the mesh pairs are very different. NPT produces very flat legs and wrong direction faces.

between the ground truth and the predicted results. The quantitative results are shown in Table 1. Our *3D-CoreNet* outperforms other methods in all metrics over two datasets. When doing pose transfer on animal data that contains more different identities, our method has more advantages.

Table 2: **Ablation study.** We use all 3 measurements here. For them, the lower is better. The PMD and CD are in units of $10^{-3}$ and the EMD is in units of $10^{-2}$. w/o means without this component. In the third and fourth column, we only use our correspondence learning module without refinement. Corr (C) uses the correlation matrix and Corr ($\mathbf{T}_m$) uses the optimal matching matrix to learn the correspondence. In w/o ElaIN, we replace our ElaIN with SPAdaIN in [37] to compare them.

| Dataset | | Corr (**C**) | Corr ($\mathbf{T}_m$) | w/o ElaIN | w/o $\mathcal{L}_{edg}$ | Full model |
|---|---|---|---|---|---|---|
| | PMD | 0.46 | 0.44 | 0.15 | 0.14 | **0.08** |
| SMPL [23] | CD | 1.39 | 1.28 | 0.37 | 0.34 | **0.22** |
| | EMD | 3.49 | 3.42 | 2.57 | 2.28 | **1.89** |

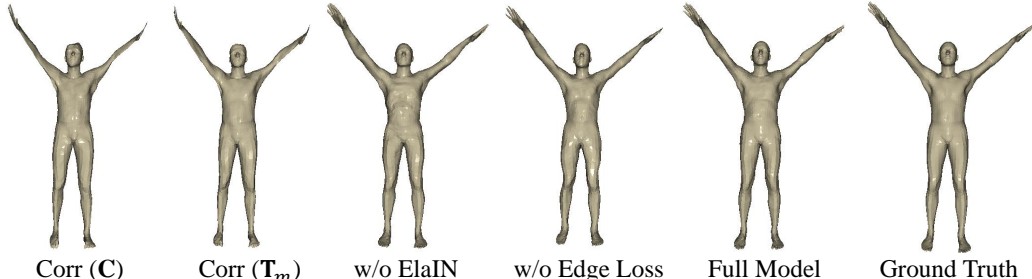

| Corr (**C**) | Corr (**T**$_m$) | w/o ElaIN | w/o Edge Loss | Full Model | Ground Truth |

Figure 5: **Ablation study results.** We test 5 variants on SMPL [23]. The first two are tested without refinement, Corr (**C**) uses the correlation matrix and Corr (**T**$_m$) uses the optimal matching matrix to learn the correspondence. The model does not perform well without refinement. Using the optimal matching matrix has a better performance than using correlation matrix. In the third column, the surface of the mesh has clear artifacts and is not smooth when we replace ElaIN with SPAdaIN.

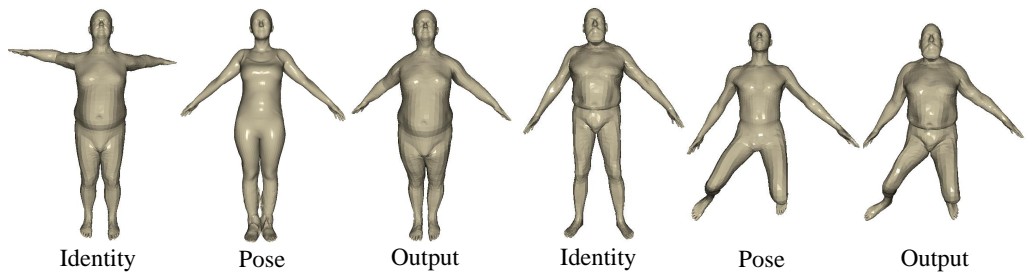

| Identity | Pose | Output | Identity | Pose | Output |

Figure 6: **Pose transfer results on human data from different datasets.** We test our model on FAUST [5] and MG-dataset [4] which contain different human meshes with SMPL [23]. Our method still has a good performance. Please refer to the supplementary material for more generated results.

## 4.2   Ablation study

In this section, we study the effectiveness of several components in our *3D-CoreNet* on human data. At first, we test our model without the refinement module. We only use our correspondence module with the correlation matrix **C** or the optimal matching matrix **T**$_m$ respectively. Here, the warped mesh will be viewed as the final output and the reconstruction loss will be calculated between the warped mesh and the ground truth. Then we will compare our ElaIN with SPAdaIN in [37] to verify the effectiveness of ElaIN. And we also test the importance of edge loss $\mathcal{L}_{edg}$.

The results are shown in Table 2 and Figure 5. We evaluate the variants with PMD, CD and EMD. As we can see, when we do not add our refinement module, the model does not perform well both qualitatively and quantitatively. And using the optimal matching matrix has a better performance than using correlation matrix. When we replace our ElaIN with SPAdaIN, the surface of the mesh has clear artifacts and is not smooth. The metrics are also worse than the full model. We can know that ElaIN is very helpful in generating high quality results. We also evaluate the importance of $\mathcal{L}_{edg}$. The connection between vertices will be better and smoother with the edge loss.

## 4.3   Generalization capability

To evaluate the generalization capability of our method, we evaluate it on FAUST [5] and MG-dataset [4] in this section. Human meshes in FAUST have the same number of vertices as SMPL [23] and have more unseen identities. In MG-dataset, the human meshes are all dressed which have 27554 vertices each and have more realistic details. As shown in Figure 6, our method can also have a good performance on FAUST and MG-dataset. In the first group, we transfer the pose from FAUST to the identity in SMPL. In the second group, we transfer the pose from SMPL to the identity in MG-dataset. Both of them transfer the pose and keep the identity successfully.

## 5 Conclusion

In this paper, we propose a correspondence-refinement network (*3D-CoreNet*) to transfer the pose of source mesh to the target mesh while retaining the identity of the target mesh. *3D-CoreNet* learns the correspondence between different meshes and refine the generated meshes jointly. Our method generates high-quality meshes with the proposed ElaIN for refinement. Compared to other methods, our model learns the correspondence without key point labeling and achieves better performance when working on both human and animal meshes. In the future, we will try to achieve the 3D pose transfer in an unsupervised manner.

## Acknowledgements

This study is supported under the RIE2020 Industry Alignment Fund - Industry Collaboration Projects (IAF-ICP) Funding Initiative, as well as cash and in-kind contribution from the industry partner. This work is supported by A*STAR through the Industry Alignment Fund - Industry Collaboration Projects Grant. This work is also supported by the National Research Foundation, Singapore under its AI Singapore Programme (AISG Award No: AISG-RP-2018-003), and the MOE Tier-1 research grants: RG28/18 (S) and RG95/20.

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
