# A    More details of 3D-CoreNet

## A.1    Network architecture

The detailed architecture of our correspondence-refinement network (*3D-CoreNet*) is shown in Table 1. We take the vertices of the identity and pose meshes as inputs. Both of them are fed into the feature extractor and the adaptive feature block. The feature extractor consists of three Conv1d-InstanceNorm-LeakyReLU blocks. Then we can calculate the optimal matching matrix with their features by solving an optimal transport problem. With the matrix, we warp the pose mesh to the coarse warped mesh. Finally, the warped mesh is better refined in the mesh refinement module with a set of elastic instance normalization residual blocks. The modulation parameters in the normalization layers are learned with elastic instance normalization.

The design of our elastic instance normalization (*ElaIN*) is shown in Figure 1. At first, we normalize the features of the warped mesh $h^i_{warp}$ with instance normalization and get the mean $\mu^i$ and standard deviation $\sigma^i$. Then, the features of the identity mesh are fed into a simple convolution layer to get $h^i_{id}$, which shares the same size with $h^i_{warp}$. We calculate the mean of $h^i_{warp}$, $h^i_{id}$ and concatenate them in the channel dimension. A fully-connected layer is employed to compute an adaptive weight $w^i$. We blend $\gamma^i$, $\sigma^i$ and $\beta^i$, $\mu^i$ elastically with $w^i$ to get the modulation parameters $\gamma'$ and $\beta'$, where $\gamma^i$ and $\beta^i$ are learned from $h^i_{id}$ with two convolution layers. Finally, we scale the normalized $h^i_{warp}$ with $\gamma'$ and shift it with $\beta'$.

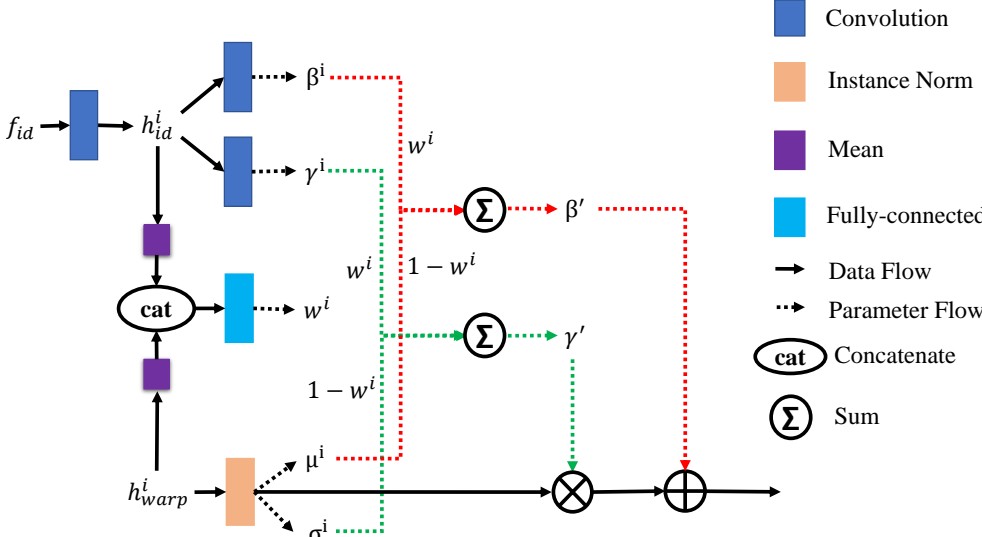

Figure 1: **The detailed design of our elastic instance normalization**. Here, we normalize the features of the warped mesh $h^i_{warp}$ with InstanceNorm and get the mean $\mu^i$ and standard deviation $\sigma^i$. Then, the features of the identity mesh are fed into a convolution layer to get $h^i_{id}$, which shares the same size with $h^i_{warp}$. We calculate the mean of $h^i_{warp}$, $h^i_{id}$ and concatenate them in channel dimension. A fully-connected layer is employed to compute an adaptive weight $w^i$. We blend $\gamma^i$, $\sigma^i$ and $\beta^i$, $\mu^i$ elastically with $w^i$ to get $\gamma'$ and $\beta'$. $\gamma^i$ and $\beta^i$ are learned from $h^i_{id}$. Finally, we scale the normalized $h^i_{warp}$ with $\gamma'$ and shift it with $\beta'$. The value on the parameter flow means the weight.

Table 1: **The network architecture of *3D-CoreNet*.** N indicates the number of vertices, D is the number of feature channels. We give an example when training on SMPL [4]. The number of vertices is 6890. $(1 \times 1)$ means the kernel size of the convolution layer is $1 \times 1$.

|  | Module | Layers | Output $(N \times D)$ |
|---|---|---|---|
| Correspondence Learning | Feature extractor | Conv1d $(1 \times 1)$ | $6890 \times 64$ |
|  |  | Conv1d $(1 \times 1)$ | $6890 \times 128$ |
|  |  | Conv1d $(1 \times 1)$ | $6890 \times 256$ |
|  | Adaptive feature block | Resblock $\times 4$ | $6890 \times 256$ |
|  |  | Conv1d $(1 \times 1)$ | $6890 \times 256$ |
|  | Optimal transport | Matching matrix | $6890 \times 6890$ |
|  | Warping | Warped mesh | $6890 \times 3$ |
| Mesh Refinement | Refinement | Conv1d $(3 \times 3)$ | $6890 \times 1024$ |
|  |  | Conv1d $(1 \times 1)$ | $6890 \times 1024$ |
|  |  | ElaIN Resblock | $6890 \times 1024$ |
|  |  | Conv1d $(1 \times 1)$ | $6890 \times 512$ |
|  |  | ElaIN Resblock | $6890 \times 512$ |
|  |  | Conv1d $(1 \times 1)$ | $6890 \times 256$ |
|  |  | ElaIN Resblock | $6890 \times 256$ |
|  |  | Conv1d $(1 \times 1)$ | $6890 \times 3$ |

## A.2 Solving OT problem with Sinkhorn algorithm

In this section, we solve the optimal transport (OT) problem defined in the main paper (Section 3.1) with Sinkhorn algorithm [5]. Following [3], we introduce an entropic regularization term to solve the OT problem efficiently,

$$
\mathbf{T}_m = \underset{\mathbf{T} \in \mathbb{R}_+^{N_{id} \times N_{pose}}}{\arg\min} \sum_{ij} \mathbf{Z}(i,j)\mathbf{T}(i,j) + \varepsilon \mathbf{T}(i,j)(\log \mathbf{T}(i,j) - 1)
$$
$$
s.t. \quad \mathbf{T} \mathbf{1}_{N_{pose}} = \mathbf{1}_{N_{id}} N_{id}^{-1}, \quad \mathbf{T}^\top \mathbf{1}_{N_{id}} = \mathbf{1}_{N_{pose}} N_{pose}^{-1}.
\tag{1}
$$

where $\mathbf{T}$, $\mathbf{Z}$ and $\mathbf{T}_m$ are the transport matrix, cost matrix and optimal matching matrix respectively, $\mathbf{1}_{N_{id}} \in \mathbb{R}^{N_{id}}$ and $\mathbf{1}_{N_{pose}} \in \mathbb{R}^{N_{pose}}$ are vectors whose elements are all 1, $\varepsilon$ is the regularization parameter. The details of the solving process are shown in Algorithm 1.

---
**Algorithm 1** Optimal transport problem with Sinkhorn algorithm.

---
**Input:** Cost matrix $\mathbf{Z}$, regularization parameter $\varepsilon$, iteration number $i_{max}$.
**Output:** Optimal matching matrix $\mathbf{T}_m$.
  $\mathbf{U} \leftarrow \exp(-\mathbf{Z}/\varepsilon)$;
  $\mathbf{a} \leftarrow \mathbf{1}_{N_{id}} N_{id}^{-1}$;
  **for** $i = 0, ..., i_{max} - 1$ **do**
    $\mathbf{b} \leftarrow (\mathbf{1}_{N_{pose}} N_{pose}^{-1})/(\mathbf{U}^\top \mathbf{a})$;
    $\mathbf{a} \leftarrow (\mathbf{1}_{N_{id}} N_{id}^{-1})/(\mathbf{U}\mathbf{b})$;
  **end for**
  $\mathbf{T}_m \leftarrow \text{diag}(\mathbf{a})\mathbf{U}\text{diag}(\mathbf{b})$.

---

### A.3 More implementation details

In Algorithm 1, we set $\varepsilon = 0.03$ and $i_{max} = 5$. $\epsilon$ in Eq. 5 is $1e$-5. We train our model on one RTX 3090 GPU. The training time is about 24 hours on the human data and about 36 hours on the animal data.

## B More experimental results

### B.1 More results on the human and animal data

In Figure 2 and Figure 3, we show more results generated by our *3D-CoreNet* on SMPL [4] and SMAL [8] respectively.

### B.2 Generalization capability

In Figure 4, we show more results generated by *3D-CoreNet* on FAUST [2] and MG-Dataset [1]. To further test the generalization capability of our model, we compare it with NPT [7]. Since DT [6] needs reference meshes as the additional inputs and there are no reference meshes when testing on the new dataset, we do not compare with DT in this section. As we can see, the results generated by our method are more smooth and realistic than NPT. The results generated by NPT always have some artifacts on the arms.

### B.3 Shape correspondence

In Figure 5 and Figure 6, we visualize the learned shape correspondence between different meshes, the vertices of the meshes are shuffled randomly before input.

### B.4 Robustness to noise

To test the robustness of our model, we add noise to the vertex ordinates of the pose mesh. The results are shown in Figure 7, our model still produces high-quality results.

### B.5 Average inference times

In this section, we compare the average inference times for every pose transfer of different methods in the same experimental settings. As shown in Table 2, the traditional deformation transfer method [6] takes the longest time compared to the deep learning-based methods. For [7], they do not learn the correspondence between meshes, so they have the shortest inference time but the generation performance is degraded. 3D-CoreNet achieves notable improvements in generating high-quality results while the inference time is also acceptable. The fourth and the fifth columns show that solving the optimal transport problem takes a very short time while improving the generation results.

Table 2: **Average inference times of different methods**. 3D-CoreNet ($\mathbf{C}$) means 3D-CoreNet with the correlation matrix and 3D-CoreNet ($\mathbf{T}_m$) means 3D-CoreNet with the optimal matching matrix.

| Method | [6] | [7] | 3D-CoreNet ($\mathbf{C}$) | 3D-CoreNet ($\mathbf{T}_m$) |
|--------|-----|-----|---------------------------|------------------------------|
| Time | 3.3352s | 0.0068s | 0.0124s | 0.0131s |

### B.6 Limitations

Although our method produces satisfactory results in most cases and has better performance than previous works, there are still some limitations that need to be solved in the future.

For example, when testing on the animal data as shown in Figure 3, the tails of the animals are difficult to handle properly (the third row and the sixth row). When evaluating our model on new identity meshes in Figure 4, if the generated mesh reveals some parts of the human body that were

Table 3: **Licenses of the assets used in this paper**.

| Data | License websites |
|------|------------------|
| SMPL [4] | https://smpl.is.tue.mpg.de/modellicense |
| SMAL [8] | https://smal.is.tue.mpg.de/license |
| MG-Dataset [1] | https://github.com/bharat-b7/MultiGarmentNetwork |
| FAUST [2] | http://faust.is.tue.mpg.de/data_license |

not exposed in the original identity mesh, such as the underarms, it will produce some artifacts (the fourth row).

## C  Licenses of the assets

The licenses of the assets used in this paper are shown in Table 3. Their licenses are given in the websites.

## D  Broader impact

We propose a novel method which has potential applications in animated movies and games by generating new poses for existing shapes with less human intervention. Our research can also have a positive impact in the vision and graphics community to inspire new ideas to generate meshes efficiently. However, new meshes generated by the model could be abused to synthesize fake content, which is the negative aspect of this research. Such issues have already drawn a lot of attention from the public. And many approaches have been proposed to solve them technologically and legally.

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

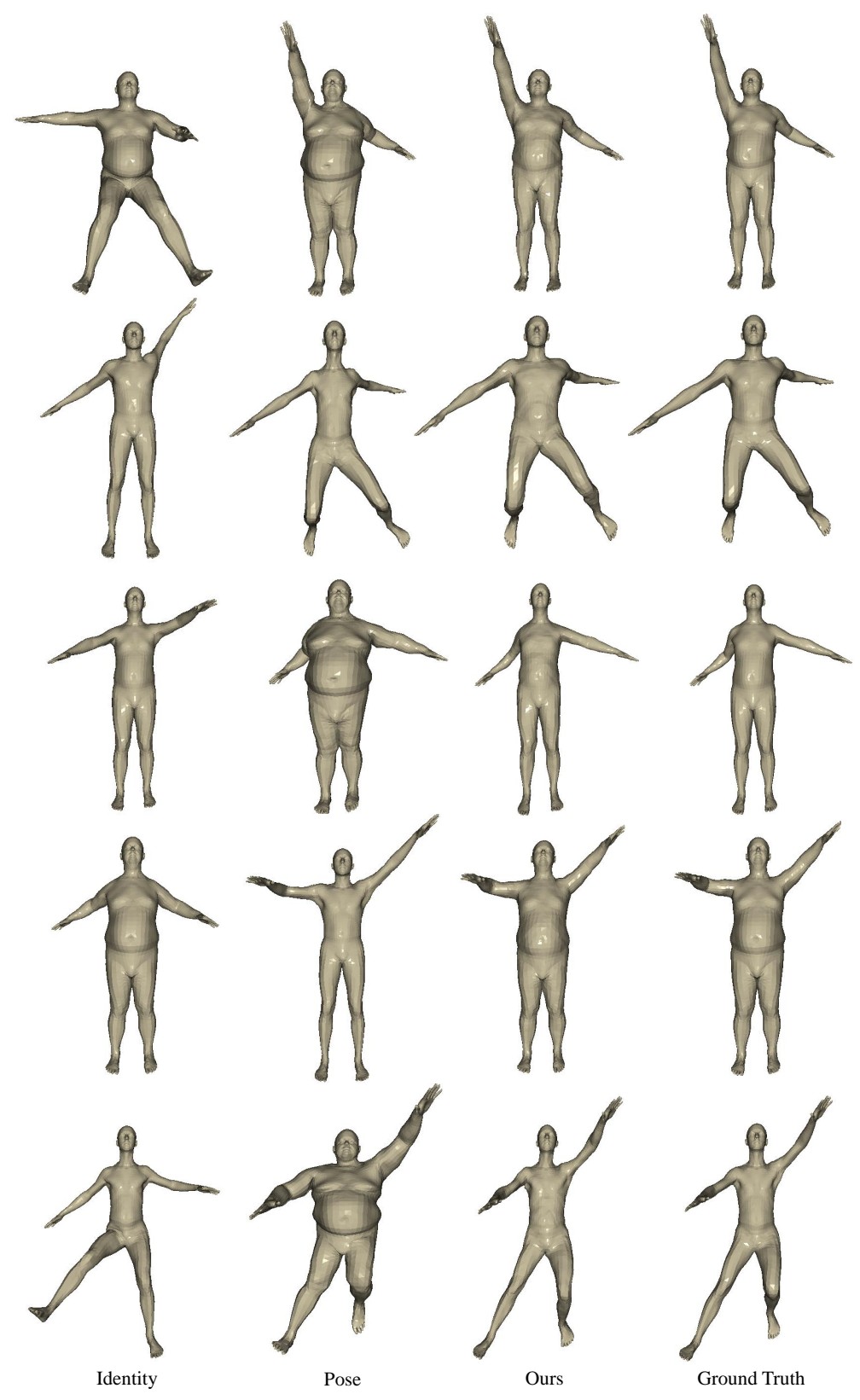

Identity       Pose       Ours       Ground Truth

Figure 2: **More results generated by *3D-CoreNet* on SMPL [4]**.

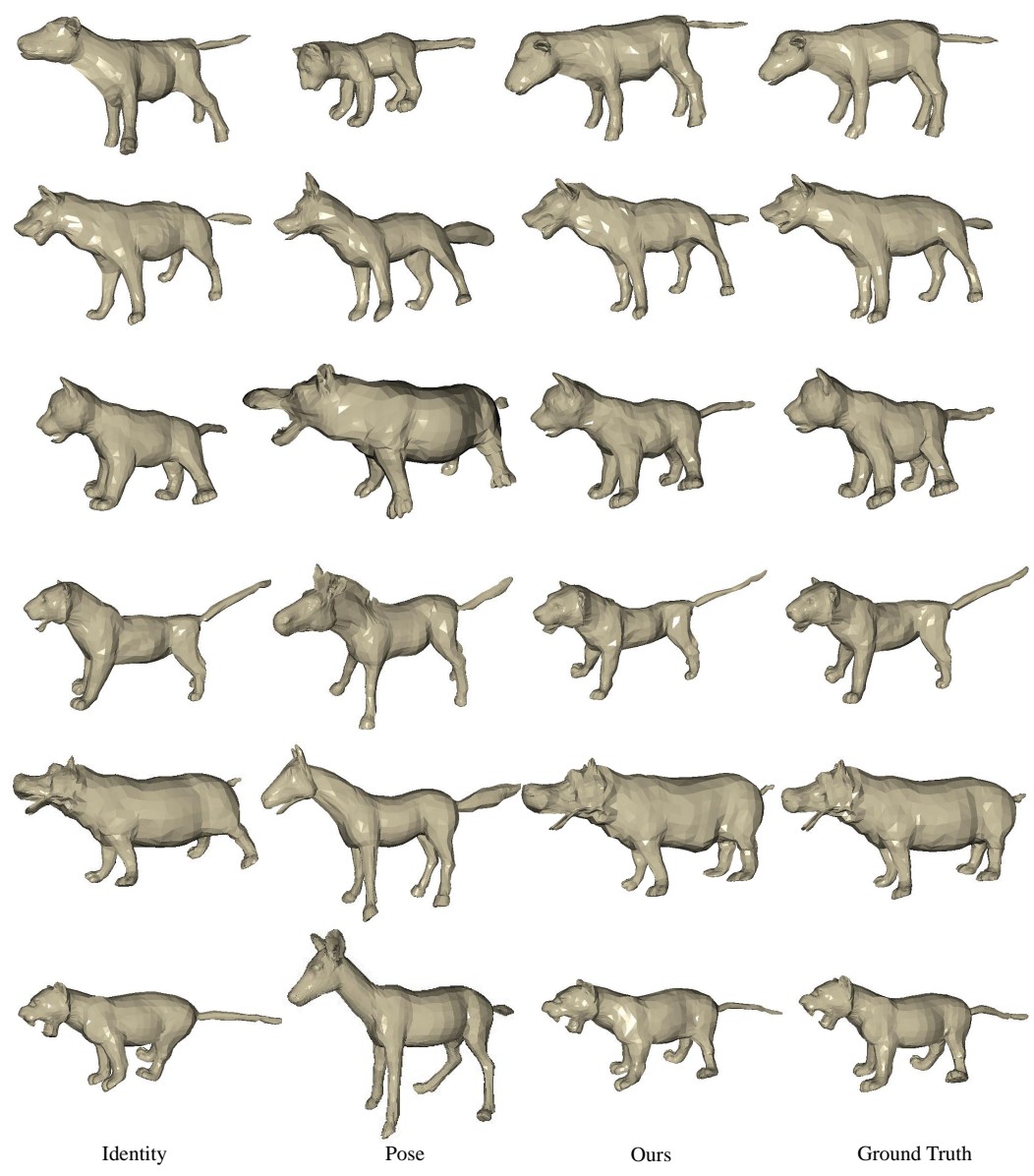

Identity     Pose     Ours     Ground Truth

Figure 3: **More results generated by *3D-CoreNet* on SMAL [8]**.

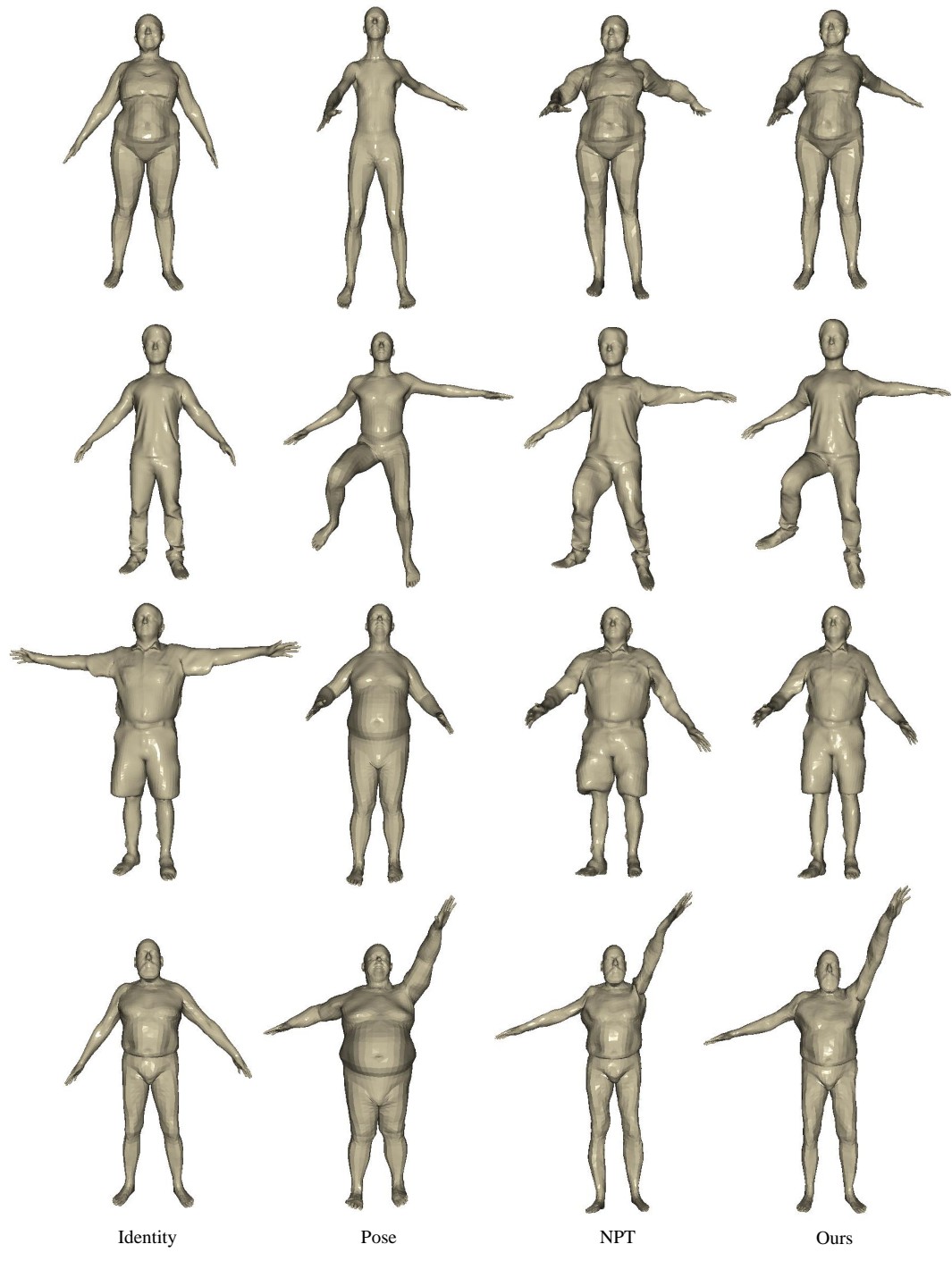

| Identity | Pose | NPT | Ours |
|----------|------|-----|------|

Figure 4: **More results generated by *3D-CoreNet* on FAUST [2] and MG-Dataset [1]**. The identity meshes are from FAUST and MG-Dataset. We compare our method with NPT [7] to test the generalization capability of our 3D-CoreNet.

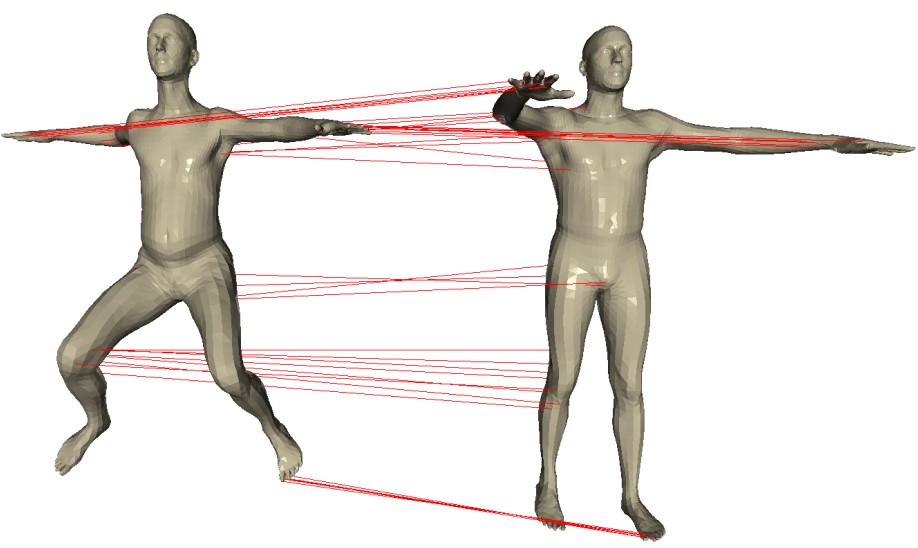

Figure 5: **Visualization of the learned correspondence between different human meshes**. We select several corresponding vertices on two human meshes to visualize. The vertices of the meshes are shuffled randomly before input.

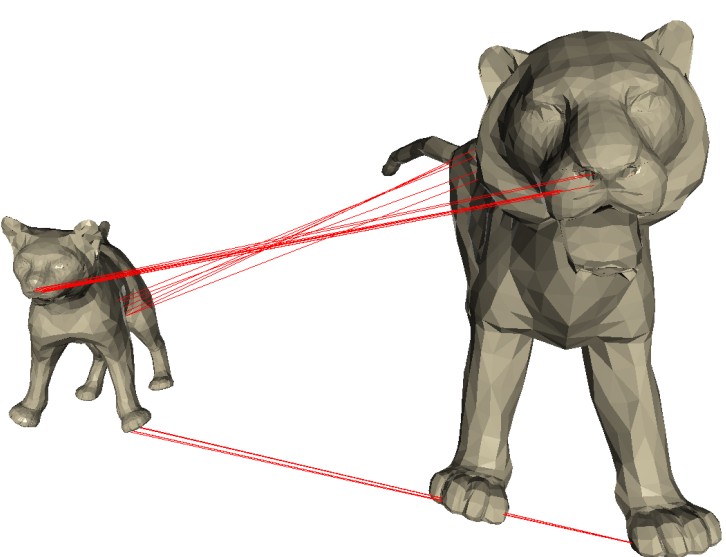

Figure 6: **Visualization of the learned correspondence between different animal meshes**. We select several corresponding vertices on two animal meshes to visualize. The vertices of the meshes are shuffled randomly before input.

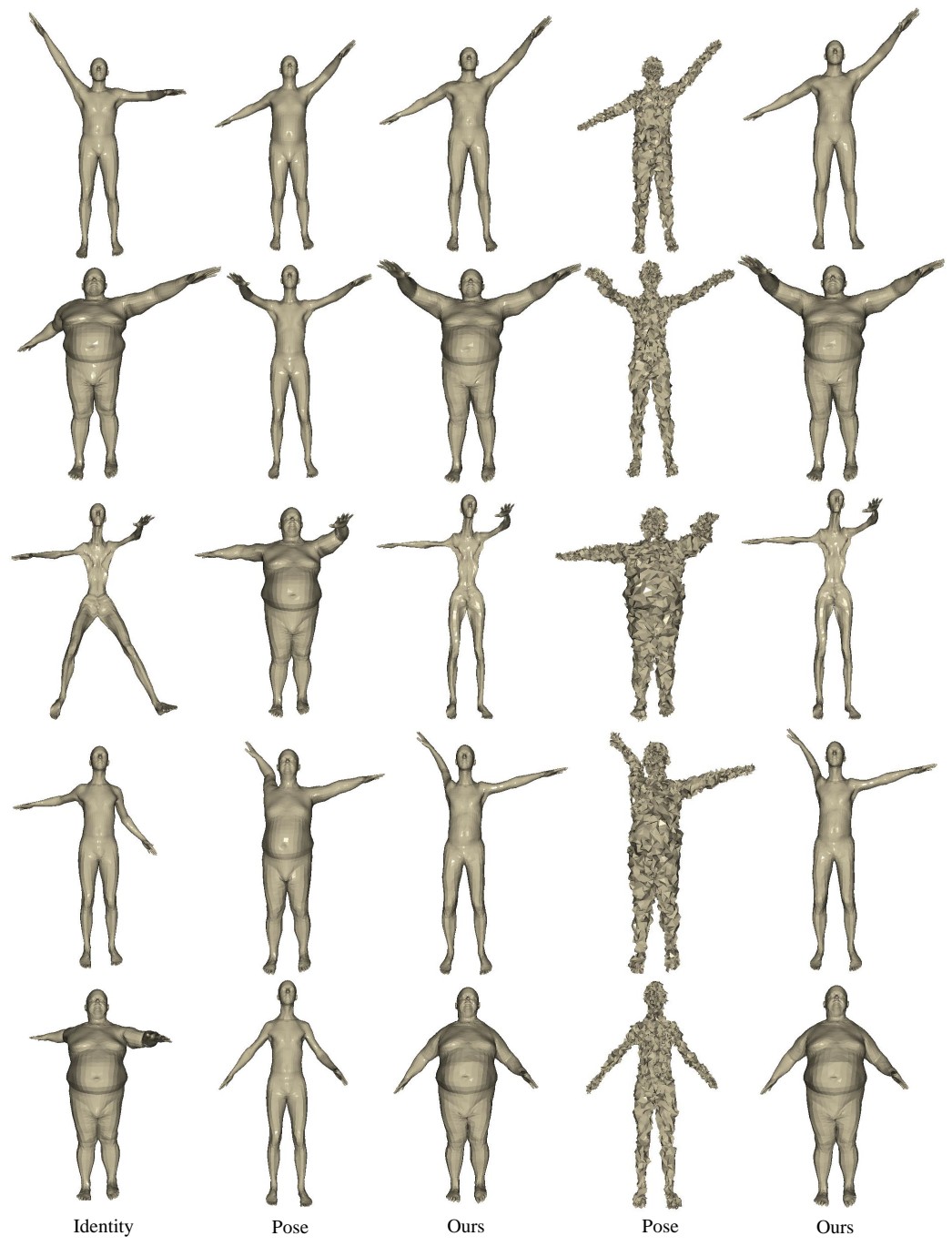

Identity      Pose      Ours      Pose      Ours

Figure 7: **Robustness to noise**. Here, we add noise to the pose mesh. Our model can still produce high-quality results.