# OpenReview forum: "3D Pose Transfer with Correspondence Learning and Mesh Refinement"
_NeurIPS.cc/2021/Conference — NeurIPS 2021 Poster_

### Official Review · Reviewer_kGky · 2021-07-16

**Rating:** 6
**Confidence:** 3

**Summary:**

This paper investigates transferring the pose of a source mesh to a target mesh while keeping its identity. Given a source pose mesh and a target identity mesh, the proposed method first learn the shape correspondence by solving an optimal transport problem and then refine the mesh with the proposed Elastic Instance Normalization. Both quantitative and qualitative results are strong on both human and animal mesh transfer.


**Limitations And Societal Impact:**

Though the results of the proposed method shown in the paper and supplementary material are quite well, adding discussions about the runtime of the 3D-CoreNet should strengthen the paper. For other concerns, please refer to the Main Review.

**Main Review:**

The paper proposes a COrrespondence-REfinement Network (3D-CoreNet) for solving the pose transfer problem of both the human and animal meshes. Similar to [37], the proposed method does not need correspondence annotations of the inputs. In 3D-CoreNet, the network uses the optimal matching matrix instead of the correlation matrix to learn the correspondence. After obtaining a coarse warped mesh, the network further refines the warped mesh with the proposed Elastic Instance Normalization (ElaIN), which is shown to be more effective than the SPAdaIN in [37].

Overall, this paper includes new technical contributions for mesh pose transfer with comprehensive experiments and convincing results. Other than that, the proposed method is an incremental update over previous work [33,37]. Some aspects also need further improvements.

- The paper claims that ElaIN is more effective than SPAdaIN, which is also supported by experimental results in Table 2 and Figure 5. These results are evaluated on the final pipeline with the warped mesh. Adding results of directly replacing SPAdaIN with ElaIN in [37] and making a comparison with the results of [37] should be more straightforward.

- The proposed 3D-CoreNet needs to compute correspondences by minimizing the total matching cost in Eq. (2). What is the time cost compared to the alternative solution using correlation matrix and the baseline [37] without correspondence learning?

- Adding a visualization of the learned correspondences between identity and pose meshes should make the experimental results more comprehensive and convincing.

**Time Spent Reviewing:**

6

---

> ### Author Response · Authors · 2021-08-10
> **Response to Reviewer kGky**
>
> Thank you for your time and feedback. In the following, we will address your comments one by one.
>
> **Q1:Ablation study on ElaIN.** **A:** In the ablation study, we replace ElaIN with SPAdaIN in [37] based on our 3D-CoreNet backbone to verify the effectiveness of ElaIN (see Table 2 and Figure 5). To follow the reviewer's advice, we show the comparison results that replace SPAdaIN with ElaIN in [37] in the table below,
>
> | Method 	| [37] 	| [37] with ElaIN 	| 3D-CoreNet 	|
> |:------:	|:----:	|:---------------:	|:----------:	|
> |   PMD  	| 0.66 	|       0.41      	|    0.08    	|
> |   CD   	| 1.42 	|       0.87      	|    0.22    	|
> |   EMD  	| 4.22 	|       3.35      	|    1.89    	|
>
> Here, we test these three models on SMPL and evaluate them with PMD, CD and EMD. For all three metrics, smaller is better. The PMD and CD are in units of $10^{-3}$ and the EMD is in units of $10^{-2}$. From the second and the third columns, we can further verify the effectiveness of ElaIN. From the third and the fourth columns, we also show the importance of the correspondence learning module.
>
> **Q2:Inference time.** **A:** The following are the average inference times for every pose transfer in the same experimental settings,
>
> | Method |  [34] |  [37]  | 3D-CoreNet ($\\mathbf{C}$) | 3D-CoreNet ($\\mathbf{T}_{m}$)  |
> | :---------: |  :-----: | :-----: | :-------------------------: | :------------------------------------------------: |
> | Time     |  3.3352s      |      0.0068s    |    0.0124s                                |    0.0131s  |
>
> where 3D-CoreNet ($\mathbf{C}$) means 3D-CoreNet with the correlation matrix and 3D-CoreNet ($\mathbf{T}_{m}$) means 3D-CoreNet with the optimal matching Matrix. As shown in the table, the traditional deformation transfer method [34] takes the longest time compared to the deep learning-based methods. For [37], they do not learn the correspondence between meshes, so they have the shortest inference time but the generation performance is degraded. 3D-CoreNet achieves notable improvements in generating high-quality results while the inference time is also acceptable. From the fourth and the fifth columns, we can find that solving the optimal transport problem takes a very short time while improving the generation results. We will add the inference time comparison in the final version.
>
> **Q3: Visualization of correspondence.** **A:** Thank you for your advice, we will add the visualization of the learned correspondence in our final version.

---

> > ### Comment · Reviewer_kGky · 2021-09-01
> > **Post-rebuttal Comment**
> >
> > Thanks for the response from the authors. The ablation study and the newly reported inference time have resolved my previous concerns. Therefore, I will keep my original rating.

---

### Official Review · Reviewer_EYda · 2021-07-16

**Rating:** 6
**Confidence:** 4

**Summary:**

This paper tackles the problem of 3D pose transfer: given an identity mesh Mi and a posed mesh Mp, the key problem is to output a mesh M' with the same id with Mi and the same pose with Mp. The proposed method is in a coarse-to-fine manner. It firstly ultizes optimal transport to learn a coarse correspondence, with which a coarse mesh can be generated. After that, a novel "elastic instance normalization" operation is proposed for mesh refinement.

**Limitations And Societal Impact:**

No more comments.

**Main Review:**

The strengths include:
1) The motivation is very clear and the proposed algorithm is also well-designed. The coarse-to-fine strategy for 3D pose transfer makes sense.
2) The experiments well validates the effectiveness of the proposed methods.

There are still some weaknesses:
1) I am not very clear that if the proposed optimal transport formula is novel or not for correspondence learning. If it is not novel, the contribution is limited. If it is novel, I think the comparisons on correspondence learning part are lacking.
2) The primary contribution to me is the proposed novel "elastic instance normalization(ElaIN)". However, the experiments to verify the effectiveness of ElaIn is not enough. For example, the authors can replace SPAdaIn with ElaIN in [37] to show if ElaIN can also benefit the method in [37].
3) Based on the above thoughts, I think the technical novelty seems not that significant.

**Time Spent Reviewing:**

4

---

> ### Author Response · Authors · 2021-08-10
> **Response to Reviewer EYda**
>
> Thank you for your time and feedback. In the following, we will address your comments one by one.
>
> **Q1: Optimal transport.** **A:** As we describe in the related work, optimal transport has many applications, such as scene flow [28], surface registration (non-deep learning) [33], semantic correspondence [22] and etc. To our best knowledge, our work is the first to use optimal transport to learn the 3D mesh correspondence with the extracted features and then help to improve the 3D pose transfer. Therefore, we believe the correspondence learning module is novel in this work. For comparisons on the correspondence learning module in our paper, we test the correlation and optimal matching matrix by evaluating the final outputs in the ablation study (see Table 2 and Figure 5). To make the experimental results more comprehensive, we will add visualizations of the learned correspondences between identity and pose meshes in the final version.
>
> **Q2: Ablation study on ElaIN.** **A:** In the ablation study, we replace ElaIN with SPAdaIN in [37] based on our 3D-CoreNet backbone to verify the effectiveness of ElaIN (see Table 2 and Figure 5). To follow the reviewer's advice, we show the comparison results that replace SPAdaIN with ElaIN in [37] in the table below,
>
> | Method 	| [37] 	| [37] with ElaIN 	| 3D-CoreNet 	|
> |:------:	|:----:	|:---------------:	|:----------:	|
> |   PMD  	| 0.66 	|       0.41      	|    0.08    	|
> |   CD   	| 1.42 	|       0.87      	|    0.22    	|
> |   EMD  	| 4.22 	|       3.35      	|    1.89    	|
>
> Here, we test these three models on SMPL and evaluate them with PMD, CD and EMD. For all three metrics, smaller is better. The PMD and CD are in units of $10^{-3}$ and the EMD is in units of $10^{-2}$. From the second and the third columns, we can further verify the effectiveness of ElaIN. From the third and the fourth columns, we also show the importance of the correspondence learning module.

---

> > ### Comment · Reviewer_EYda · 2021-09-02
> > **Post-rebuttal comment**
> >
> > Thanks authors for the rebuttal. I addressed all of my concerns.
> >
> > I will keep my rating.

---

### Official Review · Reviewer_PpVf · 2021-07-17

**Rating:** 6
**Confidence:** 3

**Summary:**

The paper proposes a reposing mesh algorithm based on identity and posing mesh, specifically, they learn the shape correspondence by optimal transport problem, and proposes lastic instance normalization to refine the generated mesh. Thus they have satisfying mesh visualization.

**Limitations And Societal Impact:**

Lack of related work like
'continuous surface embedding'
FULLY CONVOLUTIONAL MESH AUTOENCODER USING EFFICIENT SPATIALLY VARYING KERNELS
can you discuss more differences between the submission and related works?

Line 35: ' Our method does not require the two meshes to have the same number or order of vertices.', does this apply to the same person but with different vertices (say one input mesh is simplified?)

**Main Review:**

The visualization of the paper is satisfying and the proposed contribution is working and making better results both qualitatively and quantitatively.
Why is line 142 'Z=1-C'
Can you show more of the learning correspondence in simplified mesh so that we may understand more of the semantic meaning and can understand better of the corresponce?


**Time Spent Reviewing:**

2

---

> ### Author Response · Authors · 2021-08-10
> **Response to Reviewer PpVf**
>
> Thank you for your time and feedback. In the following, we will address your comments one by one.
>
> **Q1: Why is line 142 'Z=1-C'?** **A:** As we discussed in section 3.1 about the optimal transport problem, our goal is to maximize the total correlation score $\sum_{i j}\mathbf{C}(i, j)\mathbf{T}(i, j)$ with the correlation matrix $\mathbf{C}$. Then if we define $\mathbf{Z} = 1 - \mathbf{C}$ as the cost matrix, the goal is equivalent to minimize the total matching cost as shown in Eq. 2, which forms a standard optimal transport problem that can be solved with sinkhorn algorithm.
>
> **Q2: Can you show more of the learning correspondence in simplified mesh so that we may understand more of the semantic meaning and can understand better of the correspondence?** **A:** In the final version, we will add visualizations of the learned correspondences between identity and pose meshes.
>
> **Q3: Lack of related work.** **A:** The paper recommended by the reviewer mainly proposes a non-template-specific fully convolutional mesh autoencoder and it introduces a novel convolution and (un)pooling operators learned with globally shared weights and locally varying coefficients. Our work focuses on 3D pose transfer which is a type of 3D generation, so we think there is a clear difference between these two works. We will include discussions with the mentioned paper in the final version.
>
> **Q4: Line 35: 'Our method does not require the two meshes to have the same number or order of vertices.', does this apply to the same person but with different vertices (say one input mesh is simplified?)** **A:** As shown in Figure 6 in our paper and Figure 4 in the supplement materials, our model can be applied to the identity and pose meshes (different person) with different numbers of vertices. And our model can also work when applied to the same person with different vertices (simplified). The results in the paper can well verify that our method does not require the two meshes to have the same number or order of vertices, which leads to a better application of our method to real-world problems.

---

> > ### Comment · Reviewer_PpVf · 2021-08-28
> > **post-rebuttal reviews**
> >
> > The rebuttal resolves my concerns!
> > After reading other reviewers' comments and re-read the paper, I would say it better to add the 'time vs performance' table or graph as author rebuttals to reviewer 2v6p. As when I re-read the paper I found the optimal transfer may be the main time bottleneck but it does not affect the novelty of the paper.

---

### Official Review · Reviewer_2v6p · 2021-07-17

**Rating:** 6
**Confidence:** 4

**Summary:**

This paper proposes a 3D-CoreNet  to solve the 3D pose transfer problem. The 3D-CoreNet consists of two steps: a correspondence learning step and refinement step. Firstly, the authors treat  the shape correspondences learning problem as  an optimal transport problem, which can be solved without using any correspondence annotations. A coarse warped mesh can be obtained from the learned correspondence. Then, an Elastic INstance Normalization (ElaIN) is introduced to the refinement module in order to blend statistics of original features and the learned parameters from external data. State-of-the-art performance has been achieved with the proposed 3D-CoreNet.



**Ethical Concerns:**

none.

**Limitations And Societal Impact:**

The authors have discussed the limitations and potential negative societal impact in the supplementary.

**Main Review:**

Pros:
- Performance. The proposed 3D-CoreNet achieves state-of-the-art performance.
- Clarity. The paper is well written and easy to follow.
- Technical novelty. The main contributions of this paper are two folders: 1) propose to learn correspondences for a coarse estimation such that the output meshes are aligned with the identity meshes; 2) Introduce the ElaIN from image style transfer task to the 3D pose transfer task.

Cons:
- Missing references and experimental comparison.
 1. [1] Keyang Zhou et.al. Unsupervised Shape and Pose Disentanglement for 3D Meshes. ECCV2020.
 2. [2] Wang Yifan et.al.  Neural Cages for Detail-Preserving 3D Deformations. CVPR2020.
- Evaluation dataset. The authors generate their own dataset for training and testing. Is it possible to directly use the dataset from previous work [37] and compare with their results?

Further comments:
- Time complexity of the proposed approach. How to decide the number of iterations in the Sinkhorn algorithm? And how does it affect the time complexity in comparison to existing works?
- Ablation study on the correspondence learning step. One of the main contributions of this paper is the introduction of the correspondence learning step for coarse estimation. It would be better to also include the ablation study for this step (the backbone might be similar to [37] except for the ElaIN).

**Time Spent Reviewing:**

2 hours

---

> ### Author Response · Authors · 2021-08-10
> **Response to Reviewer 2v6p**
>
> Thank you for your time and feedback. In the following, we will address your comments one by one.
>
> **Q1: Missing references.** **A:** The mentioned [1] requires a dataset of meshes perfectly registered to a template for training, which means that each mesh has the same number of vertex and all the vertices are one-to-one paired between every two meshes. This is impractical in applications. In contrast, our method shuffles the mesh vertices randomly during training and testing and can be applied to meshes with different numbers of vertices, which makes it applicable to real-world problems and this is achieved by learning the correspondence.
>
> Except for the basic source meshes (identity) and target meshes (pose), the mentioned [2] requires a template source mesh as the additional input for the deformation transfer task, which is similar to [34] in our paper. They also need the corresponding landmark annotations between the template source mesh and other source meshes. Different from them, ours and [37] in our paper do not require additional input and annotations other than the basic input source meshes (identity) and target meshes (pose) for the 3D pose transfer task. Therefore, the work [34] in our paper and the mentioned [2] are designed for the same deformation transfer task, which is different from ours. And we have discussed and compared [34] with our method comprehensively in our paper, please refer to section 4.1.
>
> In summary, the experimental settings of the mentioned papers are different from ours. We will cite the mentioned papers and include detailed discussions in the final version.
>
> **Q2: Evaluation dataset.** **A:** As we discussed in line 201 in the paper, we use the same human mesh dataset for training and testing as in the previous work [37].
>
>
> **Q3: Time complexity of the proposed approach.** **A:** The iterations in the Sinkhorn algorithm are chosen empirically from multiple experiments.
>
> The following are the average inference times for every pose transfer in the same experimental settings,
>
> | Method |  [34] |  [37]  | 3D-CoreNet ($\\mathbf{C}$) | 3D-CoreNet ($\\mathbf{T}_{m}$)  |
> | :---------: |  :-----: | :-----: | :-------------------------: | :------------------------------------------------: |
> | Time     |  3.3352s      |      0.0068s    |    0.0124s                                |    0.0131s  |
>
> where 3D-CoreNet ($\\mathbf{C}$) means 3D-CoreNet with the correlation matrix and 3D-CoreNet ($\\mathbf{T}_{m}$) means 3D-CoreNet with the optimal matching matrix. As shown in the table, the traditional deformation transfer method [34] takes the longest time compared to the deep learning-based methods. For [37], they do not learn the correspondence between meshes, so they have the shortest inference time but the generation performance is degraded. 3D-CoreNet achieves notable improvements in generating high-quality results while the inference time is also acceptable. From the fourth and the fifth columns, we can find that solving the optimal transport problem takes a very short time while improving the generation results. We will add the inference time comparison in the final version.
>
>
> **Q4: Ablation study on the correspondence learning step.** **A:** Actually, when we replace ElaIN in our 3D-CoreNet with SPAdaIN in [37], the whole architecture is very close to the design of adding correspondence learning module to [37]. Therefore, you can find the ablation study on this in Figure 5 and Table 2. We will also detail this in the final version of the paper.

---

### Decision · Program_Chairs · 2021-09-27

**Decision:**

Accept (Poster)

**Comment:**

The reviewers identified a number of strengths and weaknesses in this submission. All scores are in the borderline range, but ultimately all reviewers recommend that the work should be accepted. I see not reason to overrule the reviewers in this respect and therefore recommend acceptance.